# Corrosion Behavior of Copper Exposed in Marine Tropical Atmosphere in Rapa Nui (Easter Island) Chile 20 Years after MICAT

**Rosa Vera** [1,*], **Bárbara Valverde** [1], **Elizabeth Olave** [1], **Andrés Díaz-Gómez** [1], **Rodrigo Sánchez-González** [1],
**Lisa Muñoz** [1], **Carola Martínez** [2] **and Paula Rojas** [3]

1   Instituto de Química, Facultad de Ciencias, Pontificia Universidad Católica de Valparaíso,
    Av. Universidad 330, Placilla, Valparaíso 2373223, Chile
2   Departamento de Ingeniería de Obras Civiles, Universidad de La Frontera, Francisco Salazar 01145,
    Temuco 4811230, Chile
3   Facultad de Ingeniería y Ciencias, Universidad Adolfo Ibáñez, Diagonal Las Torres 2640,
    Peñalolén 7941169, Chile
*   Correspondence: rosa.vera@pucv.cl

**Abstract:** Atmospheric corrosion of copper, exposed on a tropical island in the South-Central Pacific Ocean, was reported and compared with those of a very similar study at the same site conducted 20 years earlier. The new measurements—taken over three years of exposure, from 2010 to 2013—quantified corrosion by mass loss, characterized corrosion products by X-ray diffraction (DRX) and Raman techniques, observed the attack morphology by Scanning Electron Microscope (SEM), and evaluated the patina resistance using electrochemical techniques. The results showed a copper corrosivity category of C4, and the main copper patina compound, cuprite, was porous, nonhomogeneous, and thin. Electrochemical measurements showed cuprite layer growth as a function of the exposure time, and the morphology did not favor corrosion protection. Finally, when comparing the results to those of a study 22 years previous, the copper corrosion rates increased only slightly, even with increased contaminants associated with growing local populations and continuous tourism on the island.

**Keywords:** atmospheric corrosion; marine environments; Easter Island (Rapa Nui); corrosion maps; corrosion of copper; climate change




## 1. Introduction

Atmospheric corrosion, a worldwide concern, is largely responsible for degrading the useful life of exposed metals, structures, and materials across rural, urban, industrial, marine, and marine–industrial environments [1–7]. In 1988, the Ibero-American Map of Atmospheric Corrosiveness (MICAT, Mapa Iberoamericano de Corrosión Atmosférica) project—which set out to evaluate the behavior of carbon steel, copper, zinc, and aluminum across multiple sites—further confirmed the influence of atmospheric and meteorochemical variables on corrosion [4,5,8]. Indeed, a useful structural life is determined by variables such as relative humidity; temperature; precipitation; wind speed and direction; and atmospheric pollution (environmental chloride, sulfur dioxide, and nitrogen oxides, among others) [9,10]. Studies have additionally found an explanatory variance in the angle of exposure and the height above sea level [11,12].

Carried out 20 years after the MICAT project cited above—which created local atmospheric corrosion maps in Chile [13–15], this article provides a much-needed update on copper corrosion behavior in Rapa Nui (Easter Island) and compares it with previous results (1988–1994). This site comparison focuses on Rapa Nui, an island located in the middle of the South Pacific Ocean with practically nonexistent industrial development; however, populations—as well as the number of tourists visiting the island—have grown

considerably in recent decades. Given increased ENSO intensities and climate change, this isolated ocean site is ideal for the study of atmospheric corrosion phenomena [15–17].

The Rapa Nui Island climate is considered tropical temperate [17]. Previous authors have investigated the atmospheric corrosion of some materials in coastal and coastal-tropical environments. Castañeda et al. [18] evaluated carbon steel, galvanized steel, copper, and aluminum at sites of differing meteorochemical conditions across Cuba, demonstrating that variations in relative humidity and temperature significantly influence the deposition of environmental chloride and that reduced water movement in winter decreases coastal water salinity, therefore lowering corrosion rates. Depending on the study conditions near coastal edges, corrosion aggressiveness has been found to be CX and >CX for carbon steel and copper and aluminum at, and for, galvanized steel, C4 [1,2,18–20]. In a subtropical study, copper exposed in the Canary Islands archipelago has a ranked corrosivity category higher than CX [21] and C5 in Taking Taichung Harbor [22].

Additionally, Vera et al. found that the behavior of copper and patina color in coastal areas depend on chloride content, temperature, and relative humidity, as well as rainfall [23]. Other authors have shown that, as a function of exposure time, the atmospheric corrosion of copper in areas with salinity levels between 110 mg m$^{-2}$ d$^{-1}$ and 1640 mg m$^{-2}$ d$^{-1}$ forms superficial copper chlorides ($Cu_2(OH)_3Cl$), such as atacamite and clinoatacamite, and cuprite ($Cu_2O$) in the inner layers. The coating and sealing of cuprite by basic chlorides occurs mainly in areas where moisture is retained for a longer time and salinity is higher as a result of evaporation of the aqueous phase during the drying period [24].

Generally, copper patinas are chemically and structurally complex and have inhomogeneous surfaces, degrees of porosity, and sometimes loss of adhesion, making them prone to retaining moisture and atmospheric pollutants [25–28]. Copper patinas formed centuries ago show variable uniformity and behavior against atmospheric corrosion [27,29]. The ingress of chloride ions through patina defects can cause a localized attack on the underlying copper [26].

Pan et al. [30], while reporting on copper in a simulated coastal–industrial atmosphere, with corrosion as a function of the exposure time, reported that the main corrosion products formed were $Cu_2O$, $Cu_2Cl(OH)_3$, and $Cu_4Cl_2(OH)_6$, and, in the presence of $SO_2$, mainly $Cu_4Cl_2(OH)_6$. Additionally, Mendoza et al. identified cuprite, paratacamite, posnjakite, and brochantite as copper corrosion products in the tropical climate of Cuba [31].

The present work is thus nestled in the tradition of Atmospheric Corrosion Maps in Chile and in Latin America, as well as those under similar environmental conditions, seeking to report atmospheric corrosion behaviors of copper on Rapa Nui from 2010 to 2013 and to compare them against the results obtained from 1988 to 1994.

## 2. Materials and Methods

### 2.1. Place, Sample Preparation, Installation, and Meteorological Measurements

Rapa Nui (also known as Easter Island) is subtropical, with narrow seasonal temperature variations as mediated by oceanic temperance. The annual average is 21 °C, with a gentle seasonal range of 5 °C on average between 18 °C in the Austral winter (July–September) and 23–24 °C in the Austral summer (January–March). The total annual rainfall ranges between 1100 and 1300 mm, with an average of 140 rainy days. The average seasonal variability ranges between the minima of 70–80 mm/month (November to December) and maxima of 100–130 mm/month (April–June). The seasonal variability in precipitation results from the interplay of three climate systems: the South Pacific Anticyclone (SPA), the South Pacific Convergence Zone (SPCZ), and the westerly storm tracks (Figure 1).

Concerning the study, a frame containing 99.9% pure copper samples (Figure 2A) was installed at the assay site located within the Chilean Navy Base (Lat. 27.1565, Long. 109.4383; 22 m.a.s.l; 500 m from the coastline). Standard copper specimens (10 cm × 10 cm × 0.4 cm) were all exposed at an angle of 45° and separated by plastic insulators, following ISO 9223–9226 [32–35]. Prior to atmospheric exposure, the specimens were degreased with acetone, washed with high-purity water, dried with cold air, and stored in a desic-

cator. The cleaned and dried specimens were accurately measured and massed, ASTM G1–03 [36].

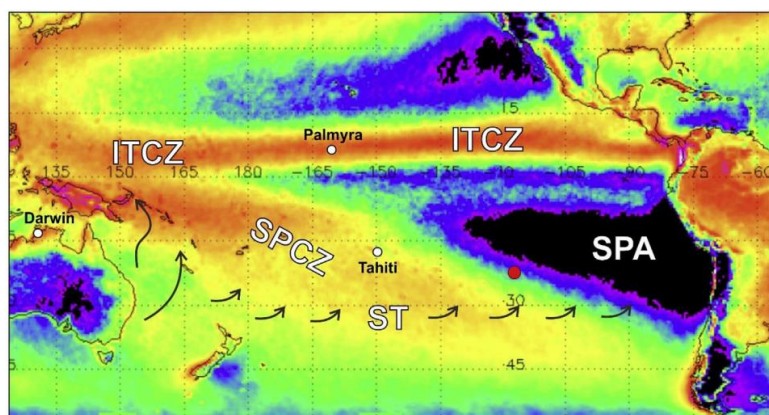

**Figure 1.** Main climate systems of the South Pacific: ITCZ, Intertropical Convergence Zone; SPCZ, South Pacific Convergence Zone; SPA, South Pacific Anticyclone; ST, Storm Tracks. Colors indicate precipitation gradients from <100 (black) to >1200 (red) mm per year, 1987–2003. Rapa Nui (Easter Island) is represented by a red dot [17].

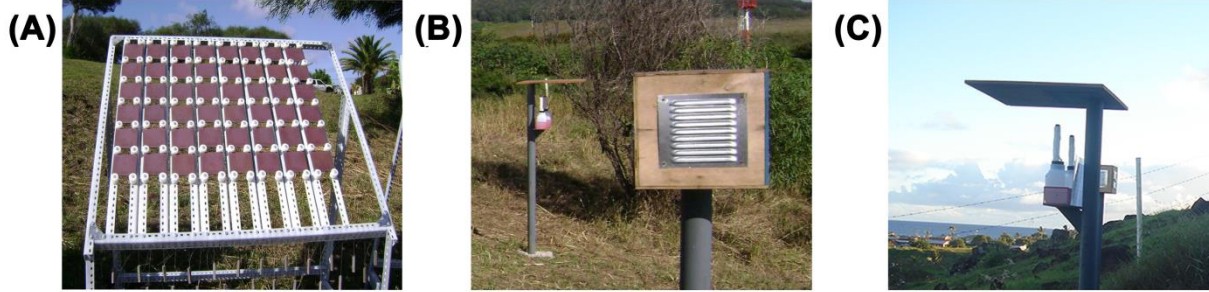

**Figure 2.** (**A**) Frame containing the copper specimens on Rapa Nui, (**B**) candles for the determination of $SO_2$, and (**C**) candles for the determination of chlorides.

Meteorological parameters recorded at each test station included the mean annual temperature (T) (°C), mean annual relative humidity (RH) (%), rainfall (mm), wind speed (m/s), and time of wetness (TOW), calculated as the number of hours for which RH ≥ 80% and T > 0 °C and expressed in hours per year. The deposition rates of atmospheric $SO_2$ (mg $SO_2$ $m^{-2}$ $d^{-1}$) and $Cl^-$ (mg $Cl^-$ $m^{-2}$ $d^{-1}$) were also calculated in accordance with ISO 9223 [32] (Figure 2B,C).

The study period was from April 2010 to April 2013 (3 years of sample exposure). Specimens were installed in the month of April, corresponding to autumn in the Southern Hemisphere.

### 2.2. Corrosion Testing

The deterioration of the material was evaluated every 3 months by measuring the mass loss in triplicate (ASTM G50) [37]. The morphology of the attack was observed under a scanning electron microscope (SEM) using a Hitachi SU 3500 with a 410-M EDAX analyzer for elemental characterization.

Corrosion products were identified using a BRUKER DRX (model D8 Advance, Karlsruhe, Germany), with Cu-Kα radiation, and a 40 KV/30 mA graphite monochromator with a scanning range of 10–70°. The corrosion products were also analyzed using Raman spectra, recorded using a Renishaw Invia Raman spectrophotometer (Great Britain) equipped with a 532-nm laser excitation source and a thermally cooled CCD camera as the detector. The resolution was set to 4 $cm^{-1}$, with 10 scans at an integration time of 10-s

recording spectra in the region of 100–4000 cm$^{-1}$ using a 50 × objective. The laser power was adjusted to 1%.

Infrared spectra (FT-IR) were recorded in a JASCO FT-IR 4600 infrared spectrophotometer (Tokyo, Japan) equipped with a DLaTGS detector. The spectral resolution in all cases was 4 cm$^{-1}$ with 32 accumulations per spectrum. The spectral range was 4000–400 cm$^{-1}$. The infrared spectra were obtained by potassium bromide (KBr) scattering.

A glow discharge optical emission spectroscopy (GDOES) depth profile analysis of the corrosion products was additionally obtained using a Spectruma Analytik GmbH GDA 750 HR (Hof, Germany) equipped with a 2.5-mm diameter anode and operating in DC excitation mode (constant voltage–constant current mode). Every sample was measured in duplicate. A glow was obtained under an argon atmosphere (5.0 quality) with an average discharge pressure of $5 \times 10^{-2}$ hPa. The excitation parameters were set to 1000 V and 12 mA, with the sputtering rate being calculated so that the measuring depth was at least 10 μm.

Finally, the protective character of the corrosion products formed on the materials was studied by, first, anodic polarization curves in 0.1 M $Na_2SO_4$ solution at 25 °C in aerated conditions with a sweep rate of 0.5 mV s$^{-1}$ using an Autolab potentiostat model 302 A, with a saturated calomel reference electrode and a platinum wire counter electrode, and second, electrochemical impedance spectroscopy measurements in an Autolab galvanostat Gamry cell with three electrodes, with copper specimen working electrodes with an exposed surface area of approximately 15 cm$^2$ in 0.1 M $Na_2SO_4$ electrolyte in aerated conditions and a high-purity (99.99%) platinum counter electrode and a saturated calomel electrode as the reference electrode. In both cases, the open circuit potential (OCP) was recorded until a steady state was reached, and impedance plots were obtained in a frequency range from 100 kHz to 10 mHz, with eight points per decade and using a sinusoidal amplitude of 10 mV peak-to-peak at E = OCP.

## 3. Results and Discussion

### 3.1. Characterization of the Test Atmosphere

Typical of oceanic islands at intermediate latitudes, the island has a tropical rainy climate (Af) that combines mild temperatures year-round, with very mild winters lacking frost or extreme cold. Its thermal regime shows an oceanic influence, with narrow daily and annual thermal oscillation. Rainfall, on the other hand, is distributed regularly throughout the year, reaching an average of 1000 mm per year. This is of convective origin, particularly in the summer season; however, during winter, the presence of some low-pressure systems brings rainfall from cold fronts. Köppen confirmed this by placing the island as having a tropical temperate with an average annual temperature of 20.5 °C between its maximum 29.7 °C in February and its minimum of 10.8 °C during August. The average relative humidity does not exceed 75%, and sunny days are around 30% of the year. The salinity of the seawater fluctuates between 35.0 and 36.4 PSU, whose variation depends on the season and the depth at which the sample is taken. As an example, the average salinity at a 4-m depth in 1999 was 35.88 PSU (T° 20.8 °C) and, at the same depth in 2015, was 35.60 PSU (T° 19.8 °C) [38].

Within this study, meteorological variables—relative humidity, temperature, wind speed, and precipitation—were recorded daily and compiled monthly. The relative humidity and temperature data together estimated the time of wetting (TOW) [39]. Figure 3 shows variations in the temperature, relative humidity, and wetting time during the three years of study, obtaining an average temperature of 20.6 °C (similar to that reported in the MICAT project), an average minimum temperature of 14.1 °C, and an average maximum temperature of 25.6 °C. The warmest months were between December and May, the least warm being between June and October. The average relative humidity was 72.7% (slightly lower than the value obtained in the MICAT project of 79.0%), with the average minimum RH 54.4% and average maximum 88.2%. With respect to the wetting time, the average throughout the 3 years of exposure was 3344 h/year, classified according to ISO 9223 [32]

in τ4. However, some daily ambient temperatures may have seen sharp increases, causing evaporation of the electrolyte and, consequently, resulted in a decreased wetting time. According to the average value calculated for TOW, the metal surface was covered by water film for about 38% of the time in relation to the total number of hours per year (8760 h/year). The exact value of the TOW reached in the MICAT project for the area under study is not available. However, the average values of the temperature and relative humidity are similar to those obtained in the present study; therefore, a τ4 classification can be estimated.

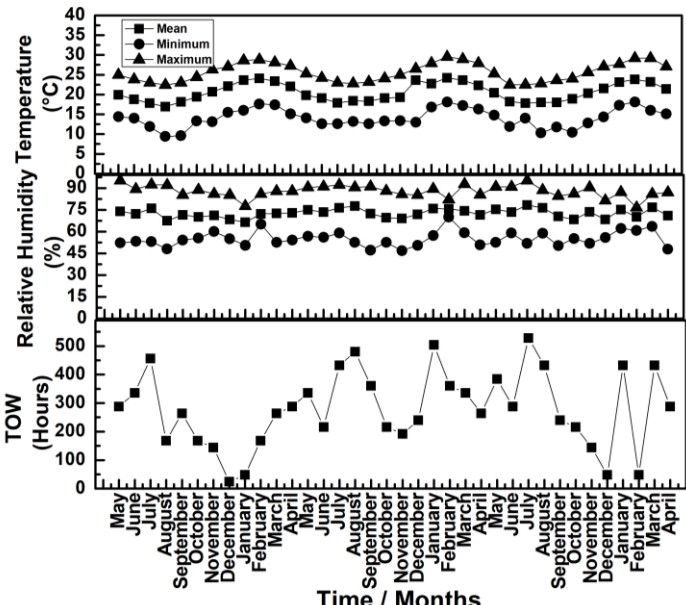

**Figure 3.** Variation of temperature, relative humidity, and wetting time during exposure (3 years).

Figure 4 shows the variation of ambient chloride (Figure 4A) and sulfur dioxide (Figure 4B) contents during the study period. The first year of exposure reached an average of 53.04 mg m$^{-2}$d$^{-1}$ for the chloride content; the second year, an average of 51.13 mg m$^{-2}$ d$^{-1}$; and the third year, an average of 48.37 mg m$^{-2}$ d$^{-1}$. All these values correspond to an S1 classification, according to ISO 9223. On the other hand, the average environmental content of sulfur dioxide for the first year was 1.89 mg m$^{-2}$ d$^{-1}$ (classification P0, rural atmosphere); in the second year, 5.41 mg m$^{-2}$ d$^{-1}$ (classification P1, urban atmosphere); and in the third, 3.30 mg m$^{-2}$ d$^{-1}$ (classification P0, rural atmosphere). When comparing the results of the environmental classification obtained in this study, with respect to the values of chloride and SO$_2$ environmental obtained on the island during the MICAT project (less than 50.0 mg m$^{-2}$ d$^{-1}$ and 1.15 mg m$^{-2}$ d$^{-1}$ (P0), respectively) (1), we can see that there is a slight increase in the chloride concentration, but the SO2 concentration turned out to be three times higher.

With the wetting time, the environmental chloride and sulfur dioxide contents could be used to estimate an environmental corrosivity category according to ISO 9223 [32]. The annual category is presented in Table 1, obtaining an environmental corrosivity category of C3, which corresponds to a medium corrosivity. This classification—generally found in coastal areas with low chloride deposition or in subtropical and tropical areas with low atmospheric pollution—was expected due to the assay site (higher than and away from the coastal edge) lacking direct contact with marine aerosol. Additionally, SO$_2$ pollution was mainly P0 (rural environment).

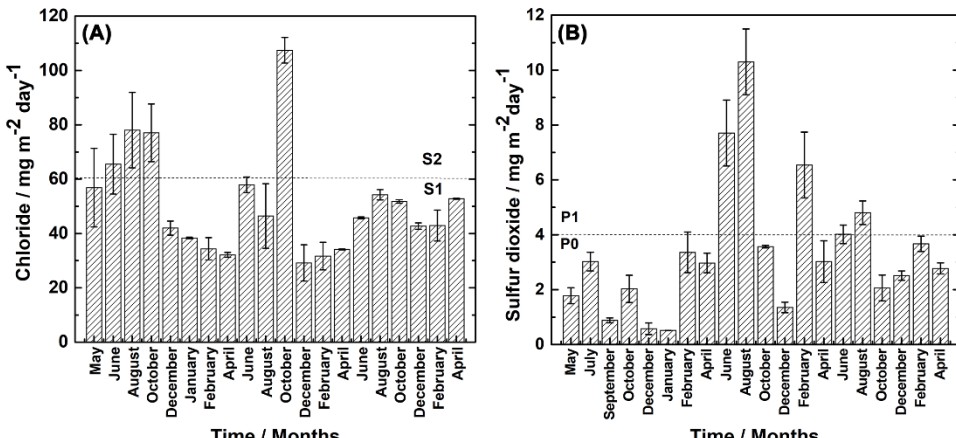

**Figure 4.** Environmental chloride (**A**) and sulfur dioxide (**B**) contents in the study period.

**Table 1.** Environmental Corrosivity Category Classification.

| Exposition (Year) | TOW ($\tau$) (hour year$^{-1}$) | Chloride (S) (mg m$^{-2}$ d$^{-1}$) | Sulfur Dioxide (P) (mg m$^{-2}$ d$^{-1}$) | Corrosiveness Category |
|---|---|---|---|---|
| 1st | 2616 | 53.04 | 1.89 | $\tau$4S1P0/C3 |
| 2nd | 3936 | 51.13 | 5.41 | $\tau$4S1P1/C3 |
| 3rd | 3480 | 48.37 | 3.30 | $\tau$4S1P0/C3 |

Considering that the time of wetting (TOW), which measures the hours in which the metal has a film of aqueous electrolyte on its surface, is determined by counting the hours that the temperature in the evaluation zone of the sample is greater than 0 °C and the relative humidity is around 70–80%. It is under these conditions that the deposition of chloride ions is favored, which, given their hygroscopicity, will maintain the wetting of the copper surface, increasing the conductivity of the film and allowing the corrosion process to occur. Therefore, an increase in the TOW will allow a greater deposition of chloride ions and therefore a greater rate of copper corrosion. However, the effect of rain and wind speed, as well as solar radiation, could generate a different relationship between the TOW and chloride ion deposition. On the other hand, the concentration of environmental $SO_2$ deposition was very small, and the classification was P0; therefore it is expected that it does not have an important influence on the copper corrosion process; rather, it will collaborate synergistically.

Figure 5A shows the amount of rainfall during the first year (760.0 mm), the second year (816.1 mm), and the third year (1028.8 mm). The distribution of rainfall throughout the year was not homogeneous. Rain may either encourage corrosion processes by increasing the TOW or reduce corrosion attacks by washing and removing contaminants on the metal surface. In the third year of exposure, the amount of rain was greater, which allowed the content of soluble chloride that was adhered to the corrosion product formed on the metal to be washed away and thus reduce the rate of progress of corrosion or increase the time of moisture favoring the corrosion process. On the other hand, Figure 5B shows the variation of the wind speed during the study time, obtaining an average between 4.4 and 4.6 m s$^{-1}$. The influence of the wind in the corrosion attack could have inhibited corrosion, dried the surface of the metallic material, and decreased the TOW; or, in contrast, it may have dragged contaminants toward exposed metal or dragged abrasive particles across the metallic surface.

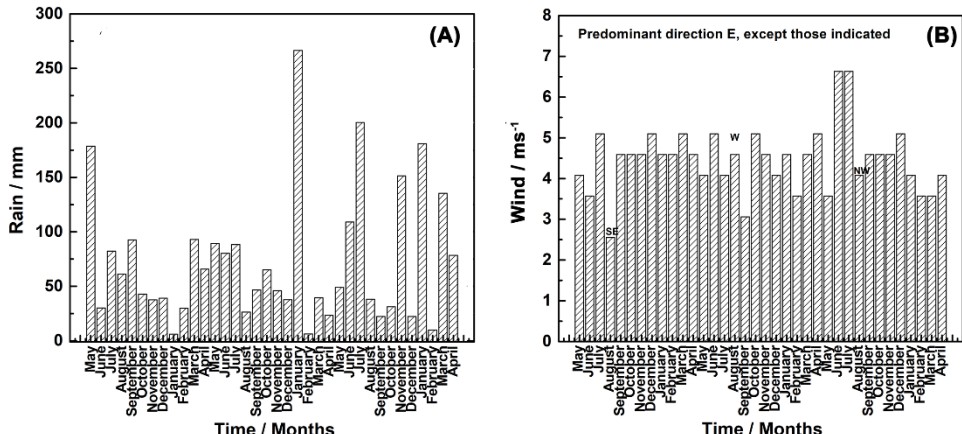

**Figure 5.** (**A**) Rainfall and (**B**) wind speed during the study period.

## 3.2. Corrosion Rate and Characteristics of Corrosion Products

Figure 6A,B shows the variations in the corrosion rate and loss of copper thickness as a function of exposure time, respectively. As seen here, as the exposure time increased, the corrosion rate decreased, and thickness loss increased. The decrease in corrosion rate can be divided into three states: rapid corrosion in the first cycle (between 3 and 6 months), which decreased the corrosion rate rapidly (approximately 35%); the second state, a deceleration in decrease of the corrosion rate, depending on the accumulation, type, and characteristics of the corrosion products protecting the surface of the metal from aggressive ions—in the case of copper, generally an internal cuprite layer ($Cu_2O$) with basic chlorides on top sealing any pores on the cuprite layer [24]; and the third, a constant corrosion rate at 1.25 $\mu m \, yr^{-1}$ after three years of exposure, which was 69% less than at three months of exposure (4.05 $\mu m \, yr^{-1}$).

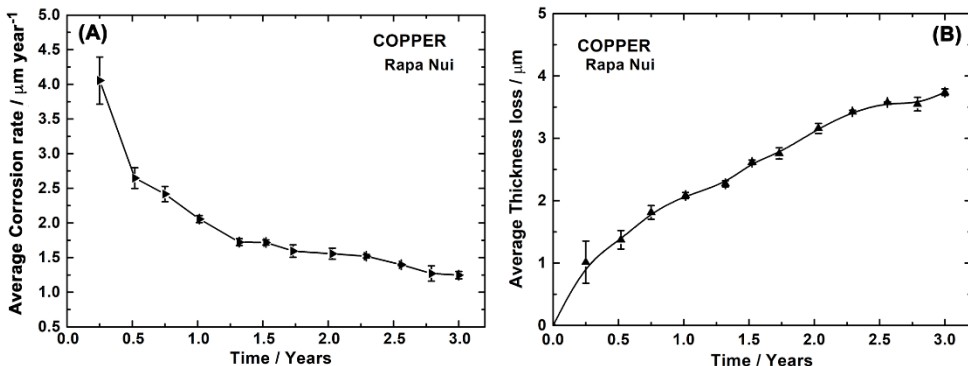

**Figure 6.** Corrosion rate (**A**) and thickness loss (**B**) of copper as a function of exposure time.

More specifically, the copper corrosion rate at one year of exposure was 2.10 $\mu m \, yr^{-1}$, which classifies as corrosivity category C4, according to ISO 9223 [32]. In this atmosphere, τ4S1P0, i.e., low average ambient chloride content (50.85 $mg \, m^{-2} \, d^{-1}$), average $SO_2$ (3.53 $mg \, m^{-2} \, d^{-1}$, and high wetting time, the behavior of copper depends mainly on the chloride concentration and wetting time. Some MICAT authors [1] mentioned that chloride concentrations above 20 $mg \, m^{-2} \, d^{-1}$ create porous corrosion product structures, which, in turn, facilitate soluble salts retention and prevent rainwater entrainment; this thus increases conductivity in the formed corrosion product layer, increasing copper surface corrosion. Notably, however, the amount of rainfall increased during exposure time in this study, reaching a total rainfall of 2605 mm after three years of exposure.

The obtained copper corrosion rate (2.10 $\mu m \, yr^{-1}$) was similar to measurements from Camet Station (Argentina) in the MICAT project [1] (2.20 $\mu m \, yr^{-1}$), which had an average

chloride content of 55.1 mg m$^{-2}$d$^{-1}$. The Rapa Nui MICAT site (300 m from the coastal edge), on the other hand, had a copper corrosion rate of 1.86 µm yr$^{-1}$ at one year of exposure, classifying it—similar to this study—as corrosivity category C4 [1]. However, that site had a reported chloride concentration less than 50 mg m$^{-2}$d$^{-1}$ (exact value not given), 1.15 mg SO$_2$ m$^{-2}$ d$^{-1}$, and no TOW value.

During the current measurement period, although the environmental SO$_2$ content tripled, its classification remained that of low contamination (P0). Given the increasing amounts of rainfall in the course of the study—and thus, contaminant flushing and TOW increase—a greater increase in the corrosion rate was not expected in comparison with the previous study (1988–1994).

The corrosion kinetics of copper under the study conditions can be described by the following empirical equation CR = At$^n$, where CR is the magnitude of corrosion (loss of thickness) of the copper at t years, A is the measure of corrosion (loss of thickness) at the first year of exposure (t = 1), n is an indicator parameter of the physicochemical behavior of the corrosion layer and its interactions with the atmosphere, and t is the exposure time of the metal in years.

The function, then, for copper thickness loss following time exposed under study conditions (Figure 6B) is given by Equation (1):

$$CR_{Cu} = 2.0763 \times t^{0.5464} \ (R^2 = 99.18\%) \tag{1}$$

where n—slightly higher than 0.5—indicates that the corrosion mechanism of copper likely corresponds to diffusion throughout the corrosion product layer. Here, the morphology and microstructure of the corrosion product film—and thus, its protective capacity—play an important role in the corrosion rate. Specifically, if the constant n is less than 1, it indicates that the corrosion products formed are protective and, for n greater than 1, an accelerated corrosion process, implying deadhesion or extreme porosity of the corrosion product film, a loss of material protection from atmospheric contaminants [40–44].

Next, the mechanism that explains the corrosion process is presented based on the results presented by several authors [24,45,46], and considering that the main compound formed corresponded to Cu$_2$O.

The dissolution of copper occurs by a redox process, according to the following reactions:
Oxidation reaction:

$$Cu \rightarrow Cu^+ + e^-$$

Reduction reaction:

$$O_2 + 2H_2O + 4\,e^- \rightarrow 4OH^-$$

Overall reaction:

$$4Cu + O_2 + 2H_2O \rightarrow 4\,Cu^+ + 4OH^-$$

Subsequently, the Cu$^+$ ion in the presence of chloride gives rise to the formation of CuCl (slightly soluble salt) and CuCl$_2$, where CuCl$_2$ by precipitation gives rise to the formation of Cu$_2$O [45].

$$2\,Cu^+ + 4Cl^- \rightarrow 2CuCl_2^-$$

$$2CuCl_2^- + 2OH^- \rightarrow Cu_2O + 2H_2O + 4Cl^-$$

As a minor product, the presence of atacamite, formed after 3 years of exposure, was identified by Raman spectroscopy. This can be attributed to the low chloride concentrations reported [45].

$$3CuCl_2^- + \frac{3}{4}O_2 + \frac{3}{2}H_2O \rightarrow Cu_2(OH)_3Cl + Cu^{2+} + 5Cl^-$$

Figure 7 shows the copper patina surfaces on both sides (e.g., differing copper sample orientations to the environment) at different exposure times. The back sides show less

influence from contaminant deposition (mainly Cl ions$^{-}$), the effect of rain by erosion, the effect of wind speed by abrasion, and the intensity of solar radiation. Indeed, more and more homogeneously distributed corrosion products were observed on the exposed face, while the nonexposed sides had a brittle corrosion product. Results that presented a similar trend were found by Wang et al. [47] when studying the corrosion of Ti coatings on brass and copper, where the greatest deterioration of the material occurred on the exposed face. After 3 months of exposure, the outward-facing corrosion product had a reddish-brown color and, after 36 months, a slight greenish color. This coloration is related to the formation of $Cu_2O$ and basic copper chlorides [45].

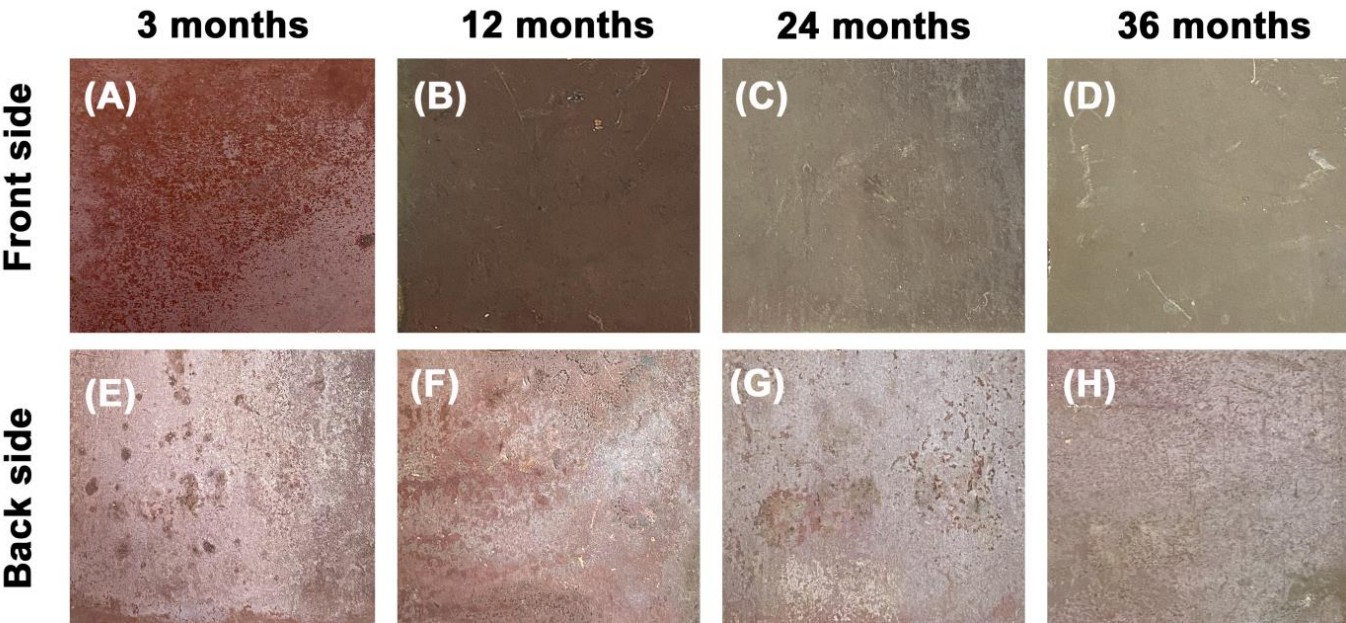

**Figure 7.** Surface appearance of the patina of copper exposed in Rapa Nui for different periods. 3 months (**A**,**E**), 12 months (**B**,**F**), 24 months (**C**,**G**), and 36 months (**D**,**H**).

Surface photographs (SEM) of the copper at 3, 12, 24, and 36 months of exposure are shown in Figure 8. The exposed sides show a nonhomogeneous granular corrosion product, with increasingly irregular growth throughout the years of exposure. The result is noncompact, rough structures, with elevations and depressions, which prevent, to some degree, the formation of protective films. The unexposed sides show a more irregular and disorderly growth and less corrosion products than the exposed side; however, the corrosion product continues to increase as a function of exposure time. These results were expected given the low environmental pollutant content (S1P0) and the total amount of rainfall during the study period.

Table 2 presents the semi-quantitative surface analysis (EDS) of copper corrosion products from Figure 8. As the exposure times increased, the copper content decreased while the oxygen content increased—for both exposed and unexposed faces—corroborating the presence of copper oxide in the corrosion product. However, the oxygen content was always lower on the unexposed side, indicating lower corrosion product formation compared to the exposed side. The chloride content also increased for both faces with the exposure time, though at a reduced rate for the exposed face at 24 and 36 months; this is likely due to higher rainfall rinsing soluble chlorides from the corrosion product. In sum, however, the chloride content was generally lower on the unexposed side, due to the shielding of the sample from the environment.

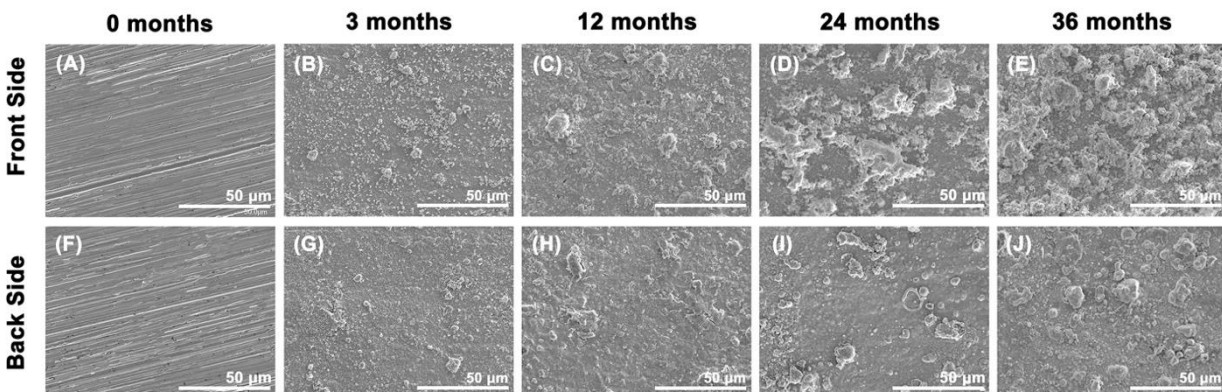

**Figure 8.** SEM of copper after 0 (**A,F**), 3 (**B,G**), 12 (**C,H**), 24 (**D,I**), and 36 (**E,J**) months of exposure, magnification 1000×.

**Table 2.** EDS of the copper corrosion product at different exposure times. 1000×.

| Atom % | 3 Months | | 12 Months | | 24 Months | | 36 Months | |
|---|---|---|---|---|---|---|---|---|
| | **Front Side** | **Back Side** | **Front Side** | **Back Side** | **Front Side** | **Back Side** | **Front Side** | **Back Side** |
| Copper | 69.65 | 64.74 | 47.11 | 56.23 | 34.88 | 41.41 | 27.09 | 43.38 |
| Oxygen | 32.81 | 29.01 | 47.36 | 38.61 | 54.56 | 46.76 | 62.84 | 50.53 |
| Chlorine | 1.34 | 2.45 | 5.53 | 5.16 | 10.07 | 8.08 | 10.56 | 9.86 |

However, for the 3 months of exposure, this relationship was not fulfilled due to the amount of rain that fell during that time (291.2 mm), which generates the washing of the soluble chlorides on the exposed face. It is important to note that the conditions that favor the washing of surface chlorides are that, during this exposure time, the formation of corrosion products is less, and some areas of the sample surface are still smooth.

Next, cross-sectional photographs (Figure 9) show a serrated copper surface, with broad attacks. The average corrosion product thickness increased with the exposure time, reaching $3.49 \pm 0.3$ μm at 3 months, $5.19 \pm 0.6$ μm at 12 months, $8.27 \pm 0.4$ μm at 24 months, and $9.13 \pm 0.5$ μm at 36 months. The corrosion product layer presented cracks and pores, with occasional small adhesion losses, which allowed contaminants and increased the corrosion attack surface. Indeed, in the third year of exposure, there was an irregular, rough corrosion product layer, with elevations and depressions, that showed the loss of patina adherence in some areas (Figure 9D). In short, patina morphology under these conditions does not suggest a completely protective compact film. Furthermore, and as mentioned previously, layer roughness and porosity impede soluble salt removal by rainwater, thus encouraging greater corrosion rates and lending to the C4 corrosivity category.

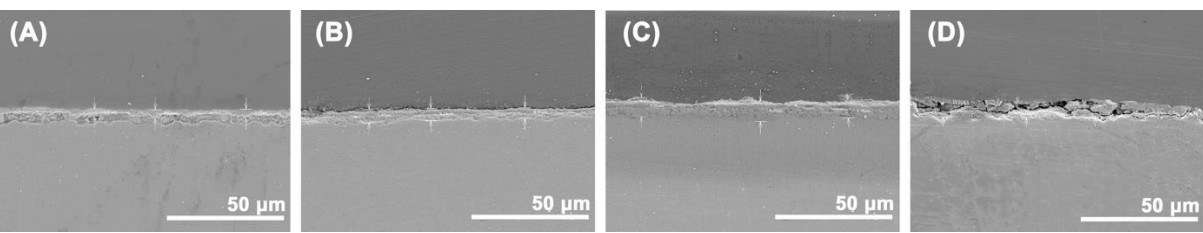

**Figure 9.** SEM of copper at different exposure times, magnification 100×: (**A**) 3 months, (**B**) 12 months, (**C**) 24 months, and (**D**) 36 months. Front side.

The GD-OES depth profiles of copper samples exposed at Rapa Nui for 1 and 3 years corroborate thin oxide films formed on the copper surface. Notably, this technique did

not find detachment or exfoliation of the corrosion product film on copper samples, even though SEM cross-section photographs (Figure 9) showed cracks in the corrosion product.

Next, X-ray diffraction determined the main copper corrosion product to be cuprite ($Cu_2O$). This is in line with Lopesino et al. [24], who found that the inner corrosion product layer in coastal zones was cuprite, as well as with Pan et al. [30], who found inner layers of cuprite corrosion in simulated coastal–industrial atmospheres, as well as in the atmospheric corrosion of copper in the middle of $H_2S$ [48]. In a study in the Nansha Islands, China, Lu et al. [45] determined that the composition of the inner layer of the corrosion product corresponded to $Cu_2O$, with a very thin outer layer of mainly $Cu_2 Cl(OH)_3$, at a corrosion rate of 7.85 μm per year$^{-1}$ (corrosivity category CX).

Corrosion products formed on copper exposed to the environment were also identified using the Raman technique at 1 and 3 years. Bands between 140 and 650 cm$^{-1}$ were related to the presence of cuprite ($Cu_2O$), and, after three years of exposure, the IR spectrum showing three bands centered at 631, 545, and 469 cm$^{-1}$ showed it as the main corrosion product, consistent with Zhang [28]. A wide band of seven components was also observed between 1250 and 750 cm$^{-1}$, with four peaks centered around 1030, 996, 906, and 835 cm$^{-1}$. These were assigned to deformations of the (-OH) group. In turn, two vibrations at 3332 and 3447 cm$^{-1}$ were associated with deformations in (O-H); this set of signals implies small amounts of atacamite ($Cu_2Cl(OH)_3$) as a corrosion product, consistent with the exposure of the sample to a marine environment (Figure 10) [1,49] and with the slight green hue presented by the patina in the sample exposed after 3 years of exposure (Figure 7D). Literature references [45] indicated that the stability of $Cu_2O$ is inversely dependent on the chloride concentration; given the environmental chloride content of the study site in Rapa Nui is 58 mg m$^{-2}$ d$^{-1}$ (category S1), the formation of cuprite was expected as the main corrosion product.

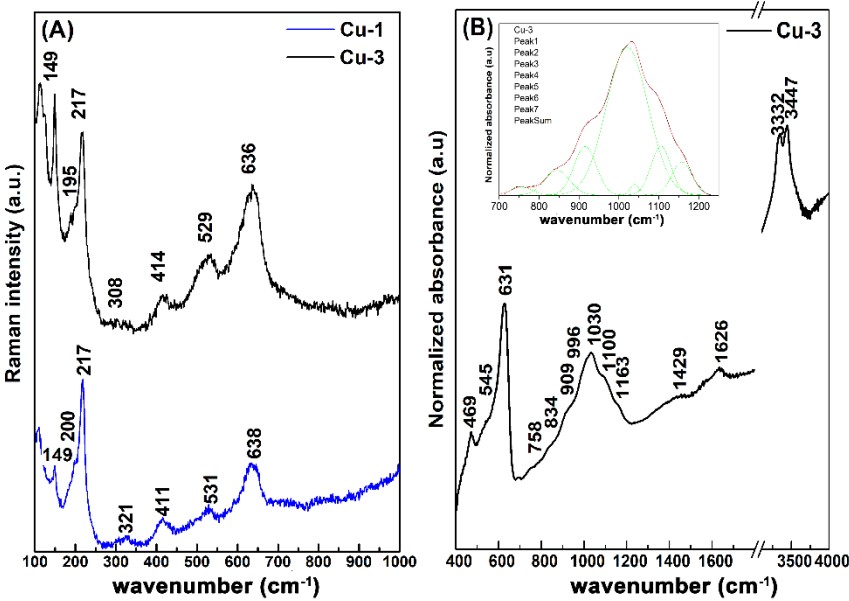

**Figure 10.** (**A**) Raman spectra of the copper samples at 1 and 3 years, recorded with a 532-nm laser line. (**B**) FTIR spectra of the corrosion product extracted from the sample at 3 years of exposure, recorded in KBr.

### 3.3. Electrochemical Measurements

The protective capability of corrosion products on the copper was analyzed by plotting the respective anodic curves in a 0.1 M $Na_2SO_4$ solution for unexposed and environmentally exposed probes (back and front sides) after one and three years. Figure 11 shows a general increase in corrosion potential and a decrease in active and passive dissolution zone current density as a function of exposure time. The corrosion potential of the bare

metal began at $-288.9$ mVsce and moved, after one year of exposure, to $-230.5$ mVsce, increasing by 58.4 mVsce. After three years, the corrosion potential was $-92.7$ mVsce; with respect to unexposed copper, an increase of 196.2 mVsce; and, to copper corrosion products after one year, of 137.8 mVsce. The increased corrosion potential corroborated the formation of a corrosion product layer that increased in thickness as a function of exposure time (Figure 9).

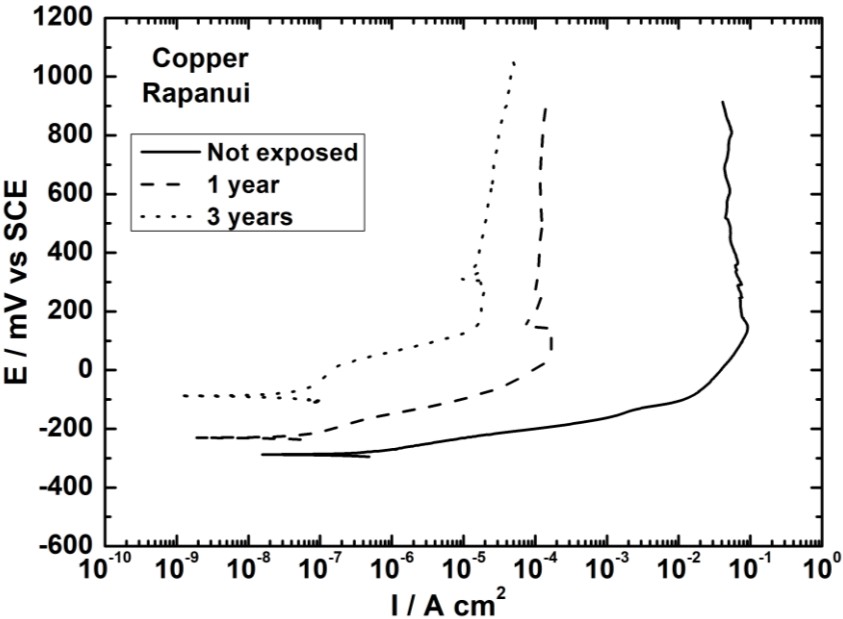

**Figure 11.** Anodic curves for unexposed copper at 1 and 3 years of exposure. Front side.

These variations further confirm that cuprite formation in marine environments resulted in porous, cracked, and nonuniformly distributed morphology on the surface of the metal, i.e., characteristics that reduced the protection capabilities. This allowed a corrosion attack from chloride to continue on copper, increasing the thickness of the layer; however, the Raman analysis in year three detected the presence of $Cu_2Cl(OH)_3$, which, in accordance with the results from previous studies on basic chlorides [1,45,49], may have sealed the cuprite pores and achieved more protective properties. This was indeed the case, corroborated by anode plots that showed a decrease in the current as a function of exposure time. Measurements showed an anodic current of $6.08 \times 10^{-2}$ A cm$^{-2}$ for unexposed copper, $1.13 \times 10^{-4}$ A cm$^{-2}$ for copper exposed for one year, and $1.69 \times 10^{-5}$ A cm$^{-2}$ for three years exposure time. This plot further shows that the active dissolution zone of the unexposed copper decreased with increased thickness (and, thus, protective character) of the corrosion product layer formed during the exposure time.

Figure 12 shows the Nyquist plot response in 0.1 M $Na_2SO_4$ solution of copper samples exposed to the environment after three years (front side). All samples showed non-ideal capacitive behavior at full frequencies due to their current and potential distribution over the surface [49], which revealed a constant phase element (CPE) behavior that can be represented by CPE parameters $Q_{CPE}$ and $\alpha$. Figure 12C shows the proposed equivalent circuit, where $CPE_C$ and $CPE_{dl}$ correspond to the capacitances of the oxide and the electrical double layer, respectively. In addition, Re represents the electrolyte resistance, $R_C$ corresponds to the oxide resistance, and Rct represents the resistance to the charge transfer occurring at the oxide/metal interface.

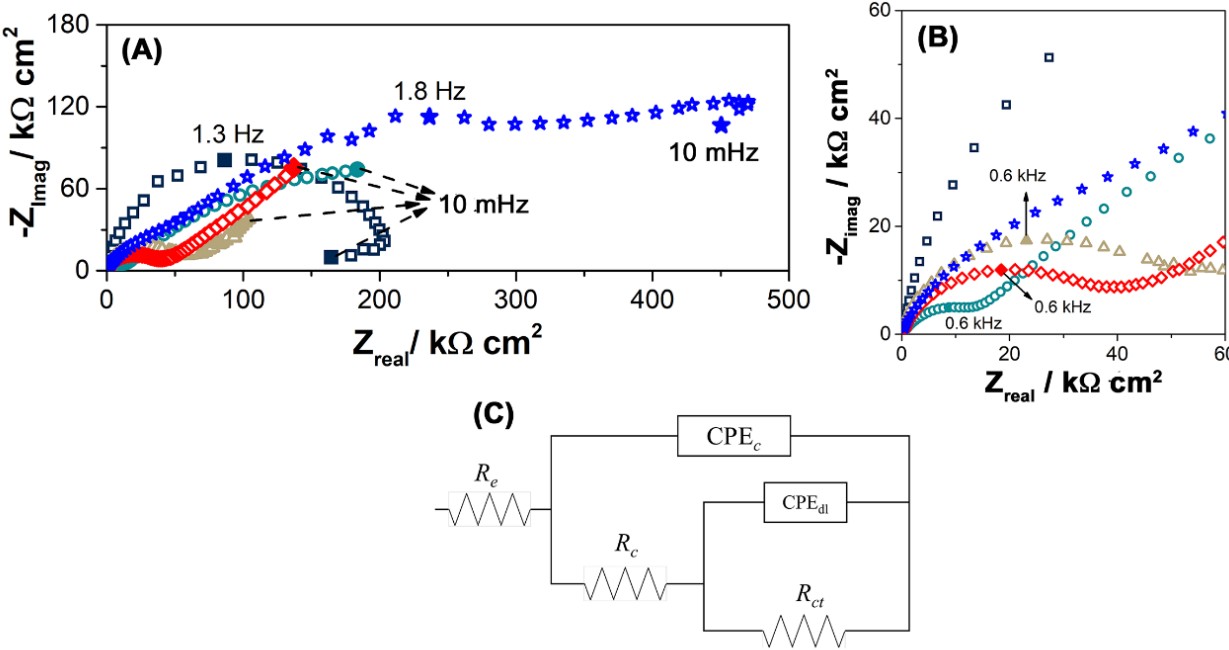

**Figure 12.** Impedance response of copper samples at different exposure times to the environment, measured in 0.1 M Na$_2$SO$_4$ solution at E = OCP. (**A**) Nyquist plot and (**B**) zoom of Nyquist plot. Exposure at 0 months (□), 3 months (○), 12 months (△), 24 months (◇), and 36 months (☆). (**C**) Equivalent circuit models.

The Bode diagram, presented in Figure 13, suggests the formation of an oxide layer, which increased with the exposure time, showing a higher modulus (Figure 13A) at 36 months of exposure. However, the decrease in the phase constant after 24 months of exposure (Figure 13B) suggests that the layer loses protectiveness with time, which could be due to a loss of homogeneity in the film.

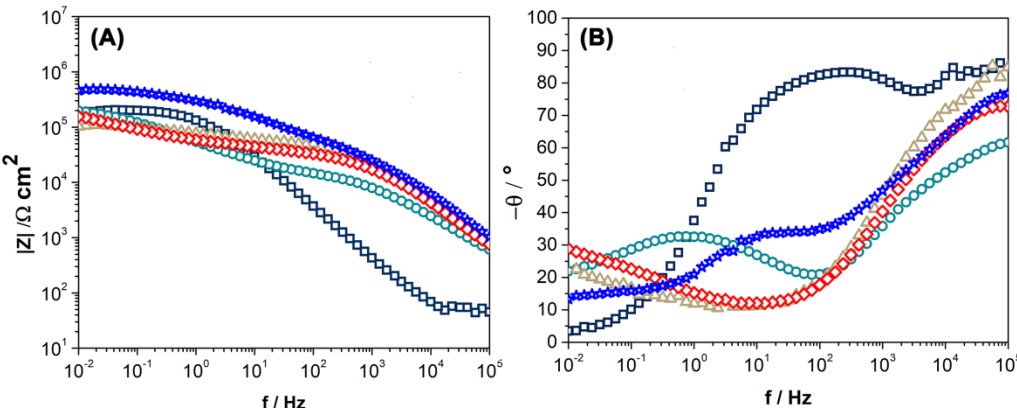

**Figure 13.** Bode diagrams of copper samples after different environmental exposure times were measured in a 0.1 M Na$_2$SO$_4$ solution at E = OCP. (**A**) Modulus and (**B**) phase were corrected by the electrolyte. Exposure at 0 months (□), 3 months (○), 12 months (△), 24 months (◇), and 36 months (☆).

The CPE parameters related to the corrosion product properties, estimated by the graphic method, are summarized in Table 3. The CPE behavior associated with the oxide film in the high frequency (HF) zones at 3 months of exposure presented a value of around 0.55 and a Q$_{CPE}$ coefficient close to $8.71 \times 10^{-7}$ (F·s$^{(\alpha-1)}$·cm$^{-2}$), which—although of a higher frequency—was very similar to that of the unexposed copper sample ($8.91 \times 10^{-7}$ (F·s$^{(\alpha-1)}$·cm$^{-2}$)). This continues to suggest the formation of the cuprite ox-

ide layer, in agreement with behaviors for the estimated constant phase element (CPE) values as the exposure time increased [50,51].

**Table 3.** Impedance parameters of copper samples at different exposure times.

| t (months) | Re ($\Omega$cm$^2$) | $-\alpha_{HF}$ | $Q_{eff,HF} \times 10^{-8}$ (FS$^{(\alpha-1)}$cm$^{-2}$) | $C_\infty \times 10^{-9}$(Fcm$^{-2}$) |
|---|---|---|---|---|
| 0 | 338 | 0.89 | 89.1 | 12.1 |
| 3 | 188 | 0.55 | 87.1 | 1.35 |
| 12 | 245 | 0.73 | 6.2 | 2.16 |
| 24 | 336 | 0.71 | 9.6 | 1.38 |
| 36 | 244 | 0.69 | 7.4 | 1.17 |

Figure 14 shows how parameter $Q_{eff}$ decreased more than tenfold after a year of exposure, which suggests an increase in the protective capacity of the oxide film formed on the surface [51]. However, as shown in Table 3, $Q_{eff}$ values denoted lost protectiveness at 2 and 3 years, which suggests that the oxide film was not homogeneous, nor compact, as in Xiao Lu et al. [45].

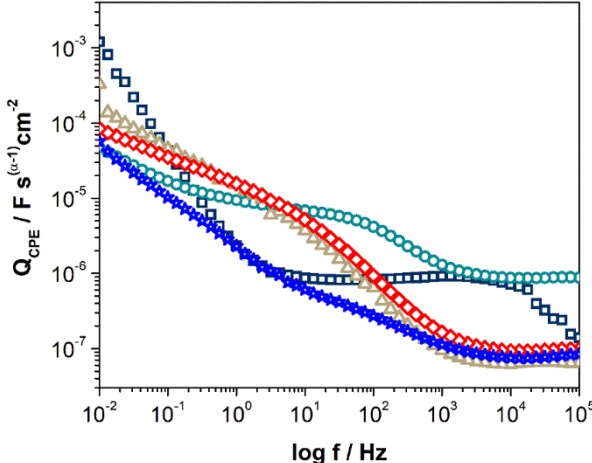

**Figure 14.** Impedance matching parameters using copper CPE parameters. Exposure at 0 months ($\square$), 3 months ($\bigcirc$), 12 months ($\triangle$), 24 months ($\diamond$), and 36 months ($\star$).

The infinite capacitances ($C_\infty$) were estimated from the Cole–Cole diagram [52,53] for copper samples exposed to the environment for up to 36 months, as shown in Figure 15. The segmented lines represent the criteria used in the determination of $C_\infty$, giving by extrapolation the real capacitance ($C_{real}$) using the slope of the complex capacitance at HF. In addition, Figure 15 shows the variation of $C_\infty$ obtained from the extrapolation line with a positive slope for all the evaluated systems, revealing that the highest value of $C_\infty$ was presented by pure copper samples before exposure, while those exposed at different times decrease by orders of magnitude. From $C_\infty$, the oxide thickness was estimated as previously described by A. S. Nguyen [53], and B. Tribollet [52].

$$C_\infty = \frac{\varepsilon \varepsilon_0}{d} \tag{2}$$

where $C_\infty$ is the capacitance in F·cm$^{-2}$, d is the oxide thickness in cm, $\varepsilon_0$ is the vacuum permittivity ($\varepsilon_0 = 8.8542 \times 10^{-14}$ F cm$^{-1}$), and $\varepsilon$ is the dielectric constant of the Copper I (Cu$_2$O) oxide layer corresponding to $\varepsilon = 7.6$ [52].

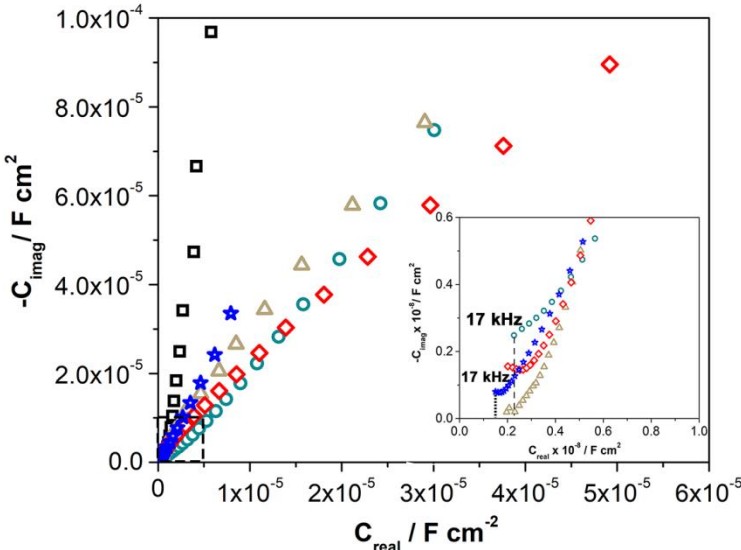

**Figure 15.** Cole–Cole plots of copper after exposure in 0.1 M Na$_2$ SO$_4$ solution at E = OCP. No exposure (□), 3 months (○), 12 months (△), 24 months (◇), and 36 months (☆).

The oxide film thickness estimations from the GD-OES, SEM, and impedance analysis after exposure to the electrolyte are summarized in Table 4. The differences are broadly attributable to technique precision, and the SEM analysis was performed locally and so may have been strongly influenced by the lack of film homogeneity. At 36 months of exposure, all techniques showed increased film thickness, independent of the irregularity or porosity presented by the copper oxide film formed (Figure 9D), as has been previously reported on typical cuprite behavior [45]. Notwithstanding, the impedance analysis suggested that the copper oxide film thickness was 3.1 < d < 5.8 µm. The thickness was almost constant up to 24 months and increased after 36 months, which may be since defects, porosity, cracks, and irregularities of the film through 12 months of exposure to 24 months were lesser, therefore favoring the migration of ions to the metal–oxide interface (Figure 9D).

**Table 4.** Variations of copper oxide thicknesses on the exposed metal surface.

| Exposure Condition, Months | 3 | 12 | 24 | 36 |
|---|---|---|---|---|
| Thickness (µm) by GD-OES | 1.6 | — | — | 6.3 |
| Thickness (µm) by SEM | 3.5 | 5.2 | 8.3 | 9.1 |
| Thickness (µm), Graphically | 4.9 | 3.1 | 4.9 | 5.8 |

## 4. Conclusions

Twenty-two years after the MICAT project on Rapa Nui Island, this study updates the behavior of copper against corrosion, considering the changes in global warming and in the contents of pollutants, and shows that the copper corrosion rate increased by 11.0% in spite of the higher elevation and greater coastal distance of the exposure zone in this study. Additionally, the environmental SO$_2$ deposition was three times higher than 22 years ago, mainly associated with the increased local population and continuous tourist presence on the island. In both study periods, the corrosion rate of exposed copper in Rapa Nui was classified in corrosivity category C4.

The main compound present in copper patinas was found to be cuprite (Cu$_2$O), with porous, nonhomogeneous, and thin morphology results similar to those obtained in research carried out 22 years ago. In this study, after 36 months of exposure, a minor amount of atacamite (Cu$_2$Cl(OH)$_3$) was also detected. The exposed sides of the copper probes presented greater, more homogeneously distributed corrosion products than on the nonexposed sides, which presented a brittle corrosion product.

The electrochemical results showed that anodic polarization curves from the front (exposed) faces have a higher corrosion potential and decreased current density in the active and passive dissolution zones as a function of the exposure time. Although this corroborates increases in the corrosion product layer thickness as a function of exposure time, this thin layer of copper oxide ($Cu_2O$) was shown to lose its protective capacity after its maximum at a year of exposure time due to increased irregularities.

**Author Contributions:** The conceptualization was developed by R.V. and P.R.; methodology, R.V., B.V., E.O. and A.D.-G.; formal analysis, R.V., L.M. and P.R.; investigation, B.V., E.O., A.D.-G., R.S.-G. and C.M.; writing—original draft preparation, R.V., L.M. and P.R.; writing—review and editing, R.V., L.M. and P.R.; supervision, R.V.; project administration, R.V. All authors have read and agreed to the published version of the manuscript.

**Funding:** The authors gratefully acknowledge the support of the Chilean government through funds from the Innova-CORFO Project No. 09CN14-5879, Fondequip EQM 160091, and the Research Department of the Pontificia Universidad Católica de Valparaíso.

**Data Availability Statement:** The raw/processed data required to reproduce these findings cannot be shared at this time due to technical or time limitations.

**Conflicts of Interest:** The authors declare no conflict of interest. The funders had no role in the design of the study; in the collection, analyses, or interpretation of the data; in the writing of the manuscript; or in the decision to publish the results.

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
