# Peer review of "Corrosion Behavior of Copper Exposed in Marine Tropical Atmosphere in Rapa Nui (Easter Island) Chile 20 Years after MICAT"

_metals, doi:10.3390/met12122082_

Round 1

Reviewer 1 Report

Comments for metals-2025430:

The paper “Corrosion behavior of copper exposed in marine tropical atmosphere in Rapa Nui (Easter Island) Chile 20 years after MICAT” investigated the atmospheric corrosion of copper exposed on a tropical island in the south-central Pacific Ocean. The corrosion products formed on the Cu surface with different exposure time were characterized by various techniques. The title is adequate and appropriate for the content of the article, the abstract is complete and suitable for inclusion by itself in an abstracting service. The conclusion represents adequately the results and main contributions of the article. But the manuscript in the current version cannot be accepted for publication. I recommend that this article be major revised. Some comments are as follows:

1.      What’s the effect of TOW, Cl and sulfur dioxide on the corrosion behavior of Cu exposed in the Easter Island environment?  A comprehensive discussion should be added to the revised manuscript.

2.      Table 2, the Cl content on the back side is slightly lower than that on the front side for all the specimens except the specimens with 3 months of exposure. Why the back side shows higher Cl content than the front side in the short exposure time?

3.      About the electrochemical tests, was the solution aerated or deaerated? Because the oxygen may have a significant on the corrosion behavior of Cu. Especially the OCP and polarization behavior are different in the solution with and without oxygen.

4.      line 407-410, the author mentioned that “the small amplitude in the frequency response suggests that the layer loses protectiveness with the passage of time, with the most protective response at one year exposure.” The above sentence is not clear. Why did the author get such a conclusion? Based on Fig. 13, the modulus of the sample with one year of exposure exhibited the lowest at low frequency. Does it mean the lower modulus relates to the higher protectiveness of the corrosion product? Some additional evidence and discussion should be added.

5.      The conclusion part is not concise and clear, it is better to rewrite this part.

Considering all the problems mentioned here above, some parts of this manuscript should be revised. I recommend that the manuscript should be major revised.

Author Response

Dear reviewer.

The entire manuscript was revised to improve the grammar and general understanding of the text. We try to comply with your recommendations and reply to their questions. We enclose PDF files with the replies for each reviewer and the revised manuscript.

Sincerely,

Rosa Vera

Reviewer 2 Report

The paper is totally fine. It can be accepted for publication after revision.

Some suggestions:

1.     The presentation of the impedance data in Nyquist plot is not completely correct, it is necessary to add some characteristics frequencies on the diagrams.

2.     Error bars of the experimental data should be given.

3.     Did observe your specimen with SEM prior to corrosion? It is better to provide some images for comparison.

4.     Maybe it is better to conduct XPS to exhibit the oxidation states of metals.

5.     It is better to cite some recent papers, e.g., https://www.nature.com/articles/s41529-022-00287-5

6.     The abstract and conclusion parts are too long. These parts should be rewritten. Not only what have done, but also the key findings and novelty.

7.     The related corrosion mechanisms should be discussed in detail.

Author Response

(The authors gave the same response as above.)

Reviewer 3 Report

This is an interesting work that provides useful and actualized data concerning the corrosivity of that zone and the corrosion resistance of the material face to this day's conditions including global warming, and pollution.
This is a well-written and presented work, technically adequate with results clearly presented. Likewise, this work is linked to work carried out at the Ibero-American level for the elaboration of corrosivity maps in different specific sites. Much of that information is no longer completely useful given climate changes, and studies to update it are very necessary.
In this work, they have only focused on one site, and even so, the change in the material's behavior is demonstrated, which is very relevant information.
In general, it is an article with the correct language use, scientific rigor, and adequate methodological procedures.  

Author Response

We greatly appreciate your excellent welcome to our work.

Thank you very much 

best regards

Rosa Vera

Round 2

Reviewer 1 Report

Comments for metals-2025430:

  The paper “Corrosion behavior of copper exposed in marine tropical atmosphere in Rapa Nui (Easter Island) Chile 20 years after MICAT” investigated the atmospheric corrosion of copper exposed on a tropical island in the south-central Pacific Ocean. The corrosion products formed on the Cu surface with different exposure time were characterized by various techniques. The title is adequate and appropriate for the content of the article, the abstract is complete and suitable for inclusion by itself in an abstracting service. The conclusion represents adequately the results and main contributions of the article. The comments of the reviewer were considered in the revised manuscript very well. The paper is recommended to accept for publication

Reviewer 2 Report

Publish it.